# A Qualitative Study on the Policy Process and Development of the National Action Plan on Antimicrobial Resistance in Singapore

**DOI:** 10.3390/antibiotics12081322

**Published:** 2023-08-16

**Authors:** Alvin Qijia Chua, Monica Verma, Karen Azupardo, Maria Margarita Lota, Li Yang Hsu, Helena Legido-Quigley

**Affiliations:** 1Saw Swee Hock School of Public Health, National University of Singapore and National University Health System, Singapore 117549, Singapore; monica.mh17@gmail.com (M.V.); mdchly@nus.edu.sg (L.Y.H.); ephhlq@nus.edu.sg (H.L.-Q.); 2College of Public Health, University of the Philippines Manila, 625 Pedro Gil St, Ermita, Manila 1000, Philippines; knazupardo@up.edu.ph (K.A.); mmlota@up.edu.ph (M.M.L.); 3The George Institute for Global Health UK, Imperial College London, 80 Wood Lane White City, London W12 0BZ, UK

**Keywords:** antimicrobial resistance, policy development, policy analysis, One Health, Singapore

## Abstract

The global public health threat of antimicrobial resistance (AMR) has been accelerated by many interrelated factors spanning across One Health—human health, animal health, and the environment. Singapore launched its own National Strategic Action Plan (NSAP) on AMR in November 2017 with the aim of tackling the growing threat of AMR in Singapore through coordinated approaches. However, little is known about the policy process and development of the NSAP in Singapore. In this study, we analysed these aspects using an AMR governance framework. In-depth interviews were conducted with 20 participants across the One Health spectrum. The interviews were transcribed verbatim and analysed thematically. Areas that were well executed included (1) good coordination across various agencies, (2) a dedicated office to coordinate the work on the NSAP, and (3) a high level of governmental support. Areas that were lacking included (1) a lack of participation from certain sectors, (2) insufficient awareness around the AMR issue, (3) constraints in information sharing, and (4) a lack of ideal indicators to track the progress in addressing AMR. Improvements in these areas will provide a more holistic One Health engagement in support of the effective planning and implementation of the NSAP.

## 1. Introduction

Antimicrobial resistance (AMR) is a slow but implacable evolutionary process that has been accelerated by human activity [1,2,3,4]. This includes interrelated factors spanning across One Health—human health, animal health, and the environment. It is estimated that approximately 1.27 million deaths were attributable to bacterial AMR in 2019, according to an assessment of the AMR burden in 204 countries [5]. To address the multifaceted global public health threat of AMR, the World Health Organization (WHO) launched the Global Action Plan (GAP) on AMR in 2015, emphasising a One Health approach for collaboration among stakeholders from various sectors, including human health, animal health, the environment, agriculture, finance, as well as informed consumers. Following the launch of the GAP, many countries also established their respective national action plans (NAPs).

Prior to the COVID-19 pandemic, AMR was among the top priorities of global public health. However, the pandemic strained health systems and deprioritised the implementation of AMR-related activities. The rampant use of antibiotics in managing COVID-19 patients was also reported, particularly during the early phase of the pandemic, which might have exacerbated the issue of AMR [6]. As COVID-19 becomes managed as an endemic disease, it is critical to prioritise AMR again and implement NAPs effectively to prevent the spread of AMR.

Singapore is a high-income city state in Asia with a diverse population of 5.64 million people [7]. The healthcare system is a hybrid of public and private services, in which public hospitals account for 80% of tertiary care, and private-sector general practitioners (GPs) cover 80% of primary care. The agricultural animal health sector is relatively small, with 3 egg-laying farms and 117 fish farms as of 2019, compared to the larger private companion, the animal health sector [8]. Sequential surveillance data showed an improvement in the prevalence rates of methicillin-resistant *Staphylococcus aureus*, *Clostridioides difficile*, and carbapenemase-producing Enterobacterales (CPE) infections in hospitals, although a significant increase in asymptomatic CPE colonisation was reported [8,9]. Low levels of multidrug-resistant Salmonella spp. were detected in local farms as well as in imported food [8]. Although significant loads of drug-resistant bacteria have been detected in waterway sites representing four different land uses, agricultural, recreational, residential, and industrial, effective water treatment systems have continued to mitigate the problem [10]. Singapore launched its own National Strategic Action Plan (NSAP) on AMR in 2017, which was developed by the country’s One Health AMR Working Group (AMRWG) to formalise existing responses, address gaps, and map future priorities across the One Health spectrum [11].

The AMRWG comprises key actors who represent each relevant agency in a government-wide effort to reduce AMR (Figure 1). The workgroup is chaired by the representative from the Ministry of Health (MOH), which oversees human health. Singapore’s Health Promotion Board (HPB) develops and implements the public education campaigns for the human health sector. The National Parks Board (NParks) manages animal health and welfare matters, while the Singapore Food Agency (SFA) is in charge of imported food sources. PUB, Singapore’s National Water Agency, looks after the water sources in Singapore, while the National Environment Agency (NEA) is responsible for the general environment. The AMRWG is supported by the AMR Coordinating Office (AMRCO) of the National Centre for Infectious Diseases (NCID).

A previous study that analysed NAPs from member states of the Association of Southeast Asian Nations (ASEAN) found areas for improvement across the NAPs, including accountability, sustained engagement, equity, behavioural economics, sustainability plans and transparency, international collaboration, as well as integration of the environmental sector [12]. However, as it was a review of the documents, we were unable to explore policy processes and discussions that stakeholders had during the NAP development and assess the actual implementation of the NAPs on the ground. Our previous publications on AMR in Singapore identified that, although a better understanding of AMR as a One Health issue was achieved, efforts were largely focused on the public hospital setting; more work was required in other areas, especially the animal agriculture and environment sectors [13,14]. However, to our knowledge, no qualitative study has been conducted to analyse the policy process and development of the NSAP of Singapore. Therefore, this study aims to address this gap using an in-depth interview (IDI) approach, guided by an AMR governance framework, and to offer policy recommendations for the way forward.

**Figure 1 antibiotics-12-01322-f001:**
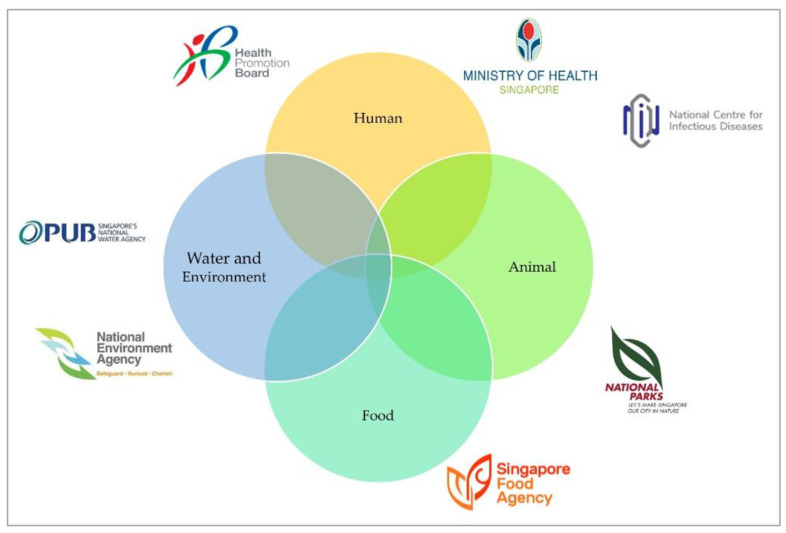
Key stakeholders of the One Health Antimicrobial Work Group, adapted from Progress Report for The National Strategic Action Plan on Antimicrobial Resistance (2018–2020) [15].

### Conceptual Framework

In our analysis of the ASEAN NAPs, we referenced the AMR governance framework published by Anderson et al. and adapted it by incorporating new domains and regrouping existing ones [16]. The updated framework now features five domains on policy design, implementation tools, monitoring and evaluation, and sustainability, with One Health engagement as the centrepiece (Figure 2). This updated framework guided our data collection and analysis [12].

## 2. Materials and Methods

### 2.1. Study Population and Data Collection

Recruitment and interviews were conducted from November 2020 to October 2021. Purposive sampling was used to recruit relevant stakeholders for AMR across the human health, animal health, and environmental sectors in Singapore. We identified relevant organisations involved in the development and execution of the NSAP. Potential participants were selected based on the organisations’ publicly available organisation charts and public directories of the Singapore government, personal contacts of the research team, or snowball recruitment from participant referrals. Potential participants were approached via email, explaining the research purpose and requesting their participation in an IDI. The invitation was deemed as rejected if no response was received after two reminder emails. We contacted 40 potential participants, of which 20 agreed to participate (Table 1). Eleven participants were female while the rest were male. Five potential participants rejected participation, stating that they were too busy to participate in view of the COVID-19 pandemic. Fifteen of them did not respond to the invitations and were deemed to have rejected participation. This could also have been due to their additional work commitments from the pandemic. The participants worked at government agencies, healthcare institutions, in academia, or private industry. Most participants were from the human health sector. We had difficulty recruiting participants from the animal health and environmental sectors. Potential participants from these sectors showed more reservation in accepting the interview invitation, perhaps based on the impression that they play a smaller role in AMR.

Interviews were conducted by up to two researchers (A.Q.C. and H.L.-Q.) in English. The interviewers were a Research Associate and an Associate Professor, both of whom had prior qualitative research training. There was no prior engagement nor established relationship between the researchers and the interviewees. Due to the COVID-19 pandemic, interviews were conducted virtually over Zoom, with only the researcher and interviewee present. Each interview lasted an hour on average and was audio-recorded. There were no repeat interviews. Interviews were based off a semi-structured interview guide, which explored our research aim in the participants’ field of expertise and interests. The interview guide (Appendix A) was constructed based on the available literature after consensus was reached among the research team [11,12,13,14,16]. Field notes were taken on reflections and interesting ideas after the interviews. No remuneration was provided to the participants.

### 2.2. Analysis

All interviews were transcribed verbatim. QSR NVivo software (Release 1.5.1) was used for data organisation and sharing among research team members. Our analysis was based on an interpretative approach, focusing on participants’ perceptions and interpretations of the topic of discussion. The AMR governance framework was used to deductively guide the coding process. Specifically, we focused on domains relevant to the policy process and development of the NSAP, including the ‘Policy design’ and ‘One Health engagement’ domains, as well as ‘Funding and resource allocation’ from the ‘Sustainability’ domain. Codes that could not fit into the framework were allowed to develop inductively as emerging themes. Coding was conducted in an iterative manner, using techniques from the grounded theory, including the constant comparative method, analysing the data line by line, labelling each line and segment of data, and using subsequent interviews to test preliminary assumptions for emerging themes [17,18]. Coding was conducted independently by two researchers (A.Q.C. and M.V.), after which discussions were conducted to resolve any disagreements and to finalise the themes. Thematic saturation was established when the research team agreed that there was sufficient and rich information for each theme and that no new themes were emerging from the data. A member check was conducted at the final stage of manuscript preparation to validate our data interpretation and to ensure that the participants’ perspectives were represented accurately. Each excerpt includes the number of the interview and sector. Findings were reported according to the COREQ checklist (Appendix A) [19].

### 2.3. Ethical Considerations

Ethical approval was obtained from the National University of Singapore, Saw Swee Hock School of Public Health Department Ethics Review Committee (SSHSPH-012). During recruitment, potential participants were provided an information sheet highlighting the research objectives and methodology. Emphasis was also placed on the confidentiality and anonymity of their responses. Before each interview, verbal consent was obtained for permission to audio-record the session and for quotes to be included anonymously in research outputs. Participants could refuse any of these options. They could also express their concerns and refuse any questions posed to them at any point during the interview. Transcripts were coded with an interview number, and any identifying data were removed from all research documents to ensure confidentiality. The research documents were password-protected, and access was restricted to the research team.

## 3. Results

We present our findings under the six main themes of participation, strategic vision, accountability and coordination, transparency, One Health engagement, as well as funding and resource allocation. These are based on relevant domains in the AMR governance framework.

### 3.1. Participation

Most participants reported a high level of participation throughout the development and implementation of the NSAP. There were regular AMRWG meetings once every two months. In addition, many participants felt that the major stakeholders were well represented and given the opportunity to contribute equally.


*“Everyone is given sufficient airtime to speak and share. I don’t think we are dominated by a particular agency. But we’re all at different stages of our work, so some agencies would be able to share more, and some agencies won’t. It’s not unusual to expect those that have more to share to be more vocal. There’s nothing to stop people from sharing what they want to.”*

*—IDI10, Animal Health*


However, contrary to the sentiment that all relevant stakeholders were well represented, one participant stated that community and primary care stakeholders were not involved and proposed reasons for it.


*“I’m afraid I may not be involved at that level. I suspect the emphasis is still in the hospitals… I’m not aware of this AMRCO. Perhaps, we will be involved when they find that the community is at stake because the spread of AMR has gone to the community… or maybe that they do not want to involve us so early yet.”*

*—IDI06, Human Health*


Similarly, a participant from the pharmaceutical sector also voiced the lack of representation from their sector.


*“It’s imperative that the people making the products to help solve the problem, have a voice… It’s like having one stakeholder who’s not at the table. We need strong partnerships from government, industry, and academia. The pharmaceutical voice would be very useful… at some point within the plan to consider financing mechanisms. There needs to be some eyes on that, just like there is in other countries.”*

*—IDI18, Human Health*


In addition to participation at the AMRWG level, participants broadly discussed engagement at three levels, political, professional, and community, for the successful development and implementation of plans.

At the political level, some participants highlighted the importance of commitment and directives from political and organisational leaders. They mentioned that international prioritisation of AMR issues politically was helpful. Some regional and global examples were shared.


*“It’s building upon the commitment from the leaders… at least at the ASEAN level, the highest level of government has already given their commitment that they would put in a lot of effort to control AMR in their countries.”*

*—IDI04, Human Health *



*“Our minister, she recently joined the Global Leaders Group on AMR early this year. So now there are quarterly meetings on those. It got everyone even more active into this despite the COVID situation.”*

* —IDI13, Environment*


Political engagement at the government and hospital management levels was discussed, highlighting the usage of research data to convince policymakers about the severity of the AMR issue. One participant mentioned that beyond contextualising the message to guide policymakers, policymakers themselves should have some basic AMR knowledge.


*“Our policymakers… I wish that there’s a course…. Then they know at least the basics of infections and AMR. I think this is important, education of policymakers.”*

* —IDI08, Human Health*


While several participants highlighted the presence of political will, governance and administrative challenges in executing the implementation plans against AMR were identified as a key barrier, as highlighted in an example from the private healthcare sector, which required more intervention from MOH.


*“Fundamentally, the chief gap that we have is governance and support from administration… it’s a different ballgame in the private sector. They will not really do something unless somebody pushes them. And that push cannot come from me… if that push comes from MOH, then maybe they will start to do something.”*

* —IDI03, Human Health*


Another participant from the pharmaceutical sector mentioned that stronger political engagement was required to sustain the antimicrobial pipeline, from research and development all the way to the patient.


*“We make medicines, and we bring them to market, but that needs to align with the government’s priorities in order to be able to get funding… Once there’s a great product, we need to fix the rest of the policy landscape to be able to bring it all the way to market, which means giving it to a patient in a hospital or someone who needs it.”*

* —IDI18, Human Health*


At the professional level, many participants expressed the importance of engaging stakeholders on the ground to drive effective implementation. One participant explained the level of engagement required for the development of legislation.


*“You need to develop the legislation together with the sector involved… It’s very important because then you work with them from the start instead of this notion that you are against them. In the end, you are trying to work with them to improve things for their production.”*

* —IDI19, Animal Health*


In the human health sector, engagement around hospital antimicrobial stewardship (AMS) programmes was most prominently shared. Some examples of on-the-ground engagement included roadshows to inform physicians about the programme and identifying AMS champions across various departments in the hospital.


*“Get the ASP team and the work that they do recognised by the hospital administrators. You have to do it top-down, at the same time, bottom-up to prepare the ground… be willing to spend more time with the ground or the prescribers on the issues about stewardship and the intention of the stewardship team… I would say engagements, a lot of engagements.”*

* —IDI02, Human Health*


In the primary care setting, some participants mentioned the engagement of large GP groups on data collection for surveillance purposes. They also talked about the education of GPs and providing resources for patient education.


*“Education to doctors in general is done by NCID. We supported them in 2019 when we developed a GP resource. This is an A5-size standee which talks about the role of antibiotics, side effects, and why you shouldn’t take antibiotics for viral infections. We have disseminated to all GPs and healthcare institutions… to support doctors in general because sometimes they are also feeling pressured… sometimes when you go to a doctor you just expect to get medication. We wanted simple infographics to help them to explain to the patients.”*

* —IDI15, Human Health*


In the animal health sector, a few participants discussed engagement with farmers and breeders to ensure adequate biosecurity through schemes and IPC programmes. They also talked about the engagement of antimicrobial wholesalers on appropriate practices in the sale of antimicrobials, as well as to obtain data from them for surveillance purposes.


*“We’re tracking it from who the wholesalers sell it to. It’s on a voluntary basis. They give us their sales data to the different animal sectors… I would think most of the wholesalers now ask for a prescription before they sell antimicrobials. They are aware of the need for these. But eventually, we will start doing more active engagement with this group of entities as the legislation and regulations kick in, you know. We will engage them as part of the drafting of these documents.”*

* —IDI10, Animal Health*


In the environmental sector, although not much was mentioned on engagement, one participant highlighted contextualising the AMR concept for their environmental engineers.


*“…Most of them are engineers, they are not well versed to know that resistant genes do not necessarily mean infectivity… every time we have genetic data, they will ask us ‘Is there a health risk?’… we’ll just try to contextualise for them.”*

* —IDI13, Environment*


Some issues on professional engagement were also raised. One participant felt that their voice was not heard and that there were groups of people who had been left out, for example, those in the media industry.


*“No one asked me for feedback on my issues… or rather I mentioned to some staff who were in charge of One Health at that time… don’t know whether he’s still around, but nothing happened.”*

* —IDI08, Human Health*



*“When we talk about education… it is still focused on public, school children or healthcare workers. We forget the reporters… up till now they don’t know what a virus or bacteria is. Very hard to talk to them, frankly speaking.”*

* —IDI08, Human Health*


At the community level, the majority of the participants discussed public engagement and ground outreach through various avenues for both human and animal health.

### 3.2. Strategic Vision

An official situational analysis was not highlighted in the NSAP; however, several participants mentioned that a similar analysis was conducted by each of the sectors.


*“We did gaps analyses prior to the publication of the action plan… what’s the current status, what are the gaps, and what we are moving forward on… the future direction.”*

* —IDI09, Animal Health*


Many participants mentioned that each edition of the NSAP should be time-bound, and, therefore, the objectives would follow the same timeline as well. One participant described how the progress was followed-up:


*“AMRCO has this Gantt chart, and it is tracked. We have a five-year work plan and my office is tracking the progress… annually we will also send it to the agencies to ask, ‘Can you help us indicate where are you along in the timeline now?’”*

* —IDI12, Human Health*


The NSAP does not have specific targets of attainment. One reason provided was the lack of baseline data to support the quantification of targets.


*“In Singapore the action plan did not have hard targets. It’s because for quite a lot of issues, we don’t even know what’s the baseline problem yet. So, we just hope to see that there is a reduction, but in the action plan there’s no target.”*

* —IDI12, Human Health*


In addition, most of the participants shared that targets for specific indicators were difficult to define. One participant highlighted that the AMR rate was one such difficult indicator because many factors contribute to it. On the other hand, indicators on the progress of implementation plans were easier to work with. One example was the increase in the number of surveillance sites.


*“For AMR, a lot of things you can’t account for it… How do you hold someone accountable for it when it’s not within their control? I think it can be a bit demoralizing for people… So unless you have some very tangible things, for example you want to achieve a 10% reduction in something very definite, like antimicrobial prescription in ARI.*

* —IDI12, Human Health*


A few participants also mentioned that the AMRWG has to start working in the next action plan concurrently while implementing the current NSAP.

### 3.3. Accountability and Coordination

Prior to the setup of the committee responsible for coordination and implementation of the NSAP, there were already pre-existing committees, such as the National Antimicrobial Resistance Control Committee and the National Antimicrobial Stewardship Expert Panel. Some participants described that these pre-existing networks were more reactive and worked on an ad hoc basis.


*“…There were formal networks to work together in areas of communicable disease control already. For example, whenever there is a zoonotic disease or food poisoning. We just leverage on that.”*

* —IDI04, Human Health*


The AMRCO was established specifically to coordinate the work of the AMRWG and efforts in implementing the NSAP. A few participants discussed how it was set up initially, as well as its role.


*“There needs to be a dedicated group of people paid to do this instead of being a side job. Or else, it’s very difficult to push things forward… that was the impetus. Also, previously when all agencies are working separately, you may lose track of each other. So that’s why AMRCO was set up to drive the agenda.”*

* —IDI12, Human Health*


Some participants also discussed the continuity of responsibility within the AMRWG to ensure that the NSAP was implemented, despite changes in the stakeholders involved. 


*“From time to time, some of these people may change, but there’s continuity because it is not driven by specific persons but driven by the organisation itself. So even if people change it’s fine.”*

* —IDI04, Human Health*


In general, most participants felt that the AMRWG and AMRCO enhanced the coordination between sectors and across different levels of each sector. They commented on how this has brought people from different sectors together, leading to further collaboration to achieve similar goals.


*“…what I could see is that it’s a lot more coordinated. And because now we know people from different agencies working on AMR, when we have issues, we are able to go back to these people… I think that since the NSAP that got all of us together, we start to see this issue not just an issue that we need to tackle within our sector. We start to see that, ‘Hey, maybe what I’m doing could help you. Or maybe what you are doing could help me. Maybe we could do something together that makes more sense than what we have been trying to do individually.’”*

* —IDI13, Environment*


One participant shared that cross-sectoral sub-working groups were formed to enhance the coordination of efforts for specific pillars of the NSAP.


*“We formed three sub-working groups when we were writing up the plan… surveillance, education, and research. These are the three areas that needed cross-sectorial discussions. Whereas optimising use, it’s either human or animal but they don’t intersect. And neither does IPC…”*

* —IDI17, Human Health*


Despite the benefits of the AMRWG and AMRCO, a few participants voiced challenges in coordination within the workgroup. For example, one participant highlighted that coordination created more work and resulted in red tape associated with interagency workflow processes.


*“What I hope to improve will be workflow or red tape issues… the accounting and finance work is quite challenging because there is no such system. We are not under a single ministry so that’s the challenge. We have to work on each other’s tempo… it cannot be that one agency says, ‘Hey, let’s go ahead and do that.’, while the others are lagging behind. We have to balance out and support each other.”*

* —IDI12, Human Health*


Collaboration outside AMRWG was also described by several participants. In human health, the workgroup works with the Ministry of Education to include AMR in school syllabi. They also coordinated with various organisations, including hospitals, public libraries, universities, malls, and public transport systems, for the annual World Antimicrobial Awareness Week. Ongoing collaboration with academia on various research projects was also reported. In animal health, there was collaboration with professional bodies and working professionals such as veterinarians to develop guidelines. There was also coordination with private companies to increase lab testing capabilities and data availability to support AMR surveillance efforts in animals. Lastly, there were discussions with the regulatory authorities on the alignment of regulatory policies for medicinal products in animal health. As for the environmental sector, there was discussion with the regulatory authorities on the list of contaminants of emerging concerns to survey at sample sites. There are also research collaborations with academia on environmental AMR projects. Despite these, one participant shared that he was unsure if there was a workgroup set up for the NSAP.


*“I know there’s an action plan but I’m not sure whether there is a task force set up. If there is a task force, it will be the work of this task force to make sure that all of us talk.”*

* —IDI02, Human Health*


Accountability, which obligates answerability to stakeholders about each other’s decisions, actions, and results, was discussed. Many participants described mechanisms, including compiled reports and progress updates of implementation plans, during workgroup meetings. When asked about the consequences of not meeting the planned targets or progress, these participants mentioned that there were generally no consequences except to review the situation and to improve on it. Most of them shared that there was self-accountability among the stakeholders.


*“For accountability… at the start of the year, they will plan what they want to achieve. If they don’t come up with it, I suppose there’s no punitive measures. But being professionals, everyone understands what they have promised. Delay sometimes is inevitable due to some other reasons or because your other work partners cannot deliver within that timeframe. So, you just have to take that… they will catch up after that.”*

* —IDI12, Human Health*


Accountability has also been described at the organisational level. For example, some participants shared their experience of accountability to senior hospital management.


*“Usually, they are accountable to the management. These things are presented to the senior management, medical boards….”*

* —IDI05, Human Health*


However, one participant shared that accountability was lacking in the hospital setting.


*“Why am I not scared that my hospital’s numbers are going up? Because no one cares… currently there’s no accountability in the hospital. You can’t say CEOs are responsible for everything. The accountability should be from MOH… It should say this person is accountable for this pathogen or this particular KPI. You are accountable for it, if not, your bonus will be cut. It should be tied to finance… I think that’s probably better than what they’re doing now.”*

* —IDI07, Human Health*


### 3.4. Transparency

Most participants reported that the NSAP and its relevant reports were publicly accessible online. However, one participant highlighted issues in dissemination:


*“I am an infectious diseases physician… I don’t know what came out of this AMR action plan…If your publicity is so good that even I don’t know, imagine others… The information that’s made is not publicly available. When I say public, I’m talking about medical professionals, infectious disease physicians who are involved in public health decision making…”*

* —IDI07, Human Health*


At a more detailed level, a few participants expressed constraints in information sharing between agencies in the AMRWG.


*“How open are we ready to share the data, I think that would be the biggest barrier… I still see some constraint in the sharing… it’s just the way the culture is.”*

* —IDI01, Environment*



*“Animal sites should tell us what they’re finding in terms of VRE in animals because we are consuming them after all. We bring in the chickens, we consume them. But they were either not monitoring or they’re very reluctant to share how much VRE they’re picking up.”*

* —IDI08, Human Health*


A participant highlighted inadequacies in the system that should be improved for more transparent information sharing:


*“…if we want to make the data sharing more transparent, the system needs to be changed. For example, the data on the sales and use of antimicrobials from farms and specific industries… it needs to be quite transparently shared. Right now, most of this information is shared through the reports.”*

* —IDI10, Animal Health*


### 3.5. One Health Engagement

All participants highlighted One Health engagement through the involvement of various sectors in the development and implementation of the NSAP. The importance of working together was emphasised by one participant.


*“I think it’s relevant and extremely important because we can’t work in silos. Whatever we’re doing in our little world, in our little, tiny niche can only contribute so much to the drama of antibiotic abuse. If the whole ecology is threatened by so many different arms, then I think every arm should be looked at… what they call the One Health concept.”*

* —IDI03, Human Health*


This was further elaborated by a participant who highlighted the need for integration.


*“You need to have a surveillance where you can detect whatever you’re looking for in the animals preferably, but at least in the food and then the humans. And then you have to be able to link the data between the food and the humans and then find out where do you want to do something about…”*

* —IDI19, Animal Health*


The set-up of AMRWG was recognised as an opportunity for agencies to contribute ideas relevant in their sector, as well as to promote a better understanding of each other’s roles and initiatives against AMR. This resulted in better synergy.


*“It is interesting because when it’s shared from another sector, sometimes you have the ‘aha!’ moments about your own sector. We start to think, ‘Oh, yeah. This target is relevant to you, then maybe do you want to look at my sector as well on this?’”*

* —IDI13, Environment*


This subsequently led to further discussions on implementation plans that considered inputs from the various sectors. One such example involved the surveillance of various water bodies:


*“Within these two years, we have quite serious discussions on surveillance of certain hotspots. But we still have to work with the other agencies to see whether our targets and sampling points makes sense from the cross-sectoral approach.”*

* —IDI13, Environment*


The development of cross-sectoral initiatives was described by many participants. Some examples include the One Health research grant and the national AMR surveillance of *Escherichia coli* across various sectors, both of which were highlighted above.

### 3.6. Funding and Resource Allocation

Funding from the government was highlighted to be of paramount importance for the sustainability of the various implementation plans, although some participants highlighted budget cuts in AMR-related work due to the COVID-19 pandemic and other competing priorities.


*“We just went through a budget-cutting exercise in view of the new wave of COVID restrictions. We are asked to tighten our belts a little bit, so the one thing that’s easiest to take away is AMR as the direct impact of the work is not as immediate.*

* —IDI11, Human Health*


Funding, specifically for the One Health work, was described, where each agency within the AMRWG contributed to supporting the work for the next five years starting in 2019:


*“There are regular meetings amongst the committee members, and we are committing a sizable sum of funding for the work to be carried out through this committee… we have put out support from the agency for the funding, so there’s a very positive outlook from our management towards this One Health work.”*

* —IDI01, Environment*


Another initiative that most participants highlighted was the One Health research fund to support cross-sectoral research specifically, although the total amount was not very sizable.

Some other examples of governmental support shared by several participants included increased funding for vaccination programmes, hospital AMS programmes, as well as national competitive and collaborative grants for AMR research. One participant voiced the need for governmental support in the pharmaceutical sector.


*“Many governments globally are exploring options for practical market level incentives to ensure that novel antibiotics are commercially sustainable. So, there’s an AMR Action Fund… all the big pharma companies put in one billion dollars in investment to bring new antibiotics to market. But again, we’ve got to work with our governments to actually get them into market.”*

* —IDI18, Human Health*


In addition to support from the government, funding from individual health institutions to support AMS and IPC programmes was mentioned.

## 4. Discussion

Our study assessed the policy process and development of the NSAP in Singapore using an AMR governance framework. We identified areas that were well executed, including (1) good coordination across various government agencies, (2) a dedicated office to coordinate the work on the NSAP, and (3) a high level of governmental support. Areas that were lacking included (1) a lack of participation from certain sectors, (2) insufficient awareness of issues around AMR, (3) constraints in information sharing, and (4) a lack of ideal indicators to track progress for the work against AMR.

The One Health Joint Plan of Action (2022–2026) emphasised that an integrated approach is crucial to tackle the problem of AMR, given that it is a One Health issue, interconnected between the human health, animal health, and environmental sectors [20]. Unfortunately, the One Health movement has been described as “a conglomeration with many different players and often uncoordinated actions” [21]. Studies in many countries expressed difficulty in cross-sectoral coordination, with many reasons, including (1) a lack of coordinated governance structures that work in tandem, (2) incompletely and insufficiently defined roles for stakeholders, and (3) differences in priorities for AMR among various sectors [22,23,24,25,26]. On the contrary, our participants highlighted good coordination across various agencies, although it resulted in increased administrative work. Part of the success was attributed to AMRCO, which was set up specifically to coordinate the work of AMRWG and efforts in implementing the NSAP.

The establishment of AMRCO was praised by our participants as being integral to enhancing cross-sectoral coordination. In addition, members of AMRCO were hired specifically for the job, a working model where Singapore is among one of the early adopters internationally. The secretariat support provided by AMRCO kept all stakeholders engaged via the organisation of regular meetings with clear action plans and ensured accountability through oversight of the implementation, monitoring, and evaluation of the NSAP. The secretariat, which involved fully committed individuals with high social capital, could form a strong social fabric with policymakers to tackle AMR effectively in a sustainable manner [26]. The analysis of the ASEAN NAPs showed that all countries had multisectoral committees or technical working groups to develop and implement the NAPs, but very few countries had dedicated manpower to coordinate efforts to combat AMR [12]. Countries, such as the Philippines, Indonesia, Myanmar, and Thailand, mentioned the formation of coordinating committees. However, these committees were formed by members who had another existing appointment in the ministries.

The AMRWG and AMRCO would not have been possible without a high level of governmental support, which is necessary for an enabling environment for AMR policies. Politicians openly supported the AMR work through public dialogue and through participation in international groups, such as the Global Leaders Group on Antimicrobial Resistance [27,28]. Monetary support through funding of various initiatives was described by the participants, which is important to ensure the sustainability of efforts to tackle AMR [29]. Unfortunately, this support was not seen downstream in certain sectors, for example, the private healthcare sector, which was described as having insufficient support from the management to implement AMR-related activities. Effective communication with policymakers can help garner their buy in for support. The Society for Public Health Education (SOPHE) suggested three rules of educating policymakers: (1) keeping messages short and simple, (2) making the information relevant to them, and (3) using stories to ensure that the messages stick [30]. In addition, SOPHE suggested action plans and education tool kits for effective education. Supportive leadership from the top is necessary to stabilise and carry initiatives forward for long-term sustainable progress. However, full implementation of One Health and AMR-related activities require both top-down and bottom-up support, as issues are often discovered, and innovative ideas are conceptualised on the ground [21]. In addition, it was shown that support for a policy put forward by a non-governmental stakeholder such as a medical scientist was stronger compared to the government [31]. This reinforces the need for a diffusion of governance involving a wider range of stakeholders, as opposed to the traditional model dominated by governmental agencies to secure long-term success [32,33].

Participation from all relevant stakeholders in the development and implementation of AMR initiatives is key, as no individual sector can solve the complex AMR problem alone. A review of 78 NAPs from around the world identified that the animal health and environment sectors were largely missing from many NAPs [34]. This resonates with the findings from other similar AMR studies conducted in Bangladesh, Malaysia, the Philippines, Tanzania, and Thailand [24,26,35,36,37]. In our previous study conducted in Singapore, we reported the need to address AMR in the animal health and environmental sectors as well [14]. Since then, there have been significant developments in the AMR implementation plans in these sectors [10,38,39]. At the international level, there was a strong emphasis to integrate the environment into the One Health approach as well. In March 2022, the environmental sector was formally included, with the extension of the Tripartite Agreement on the One Health cooperation of WHO, Food and Agriculture Organization, and World Organisation for Animal Health (founded as OIE) to a Quadripartite through partnership with the United Nations Environment Programme [40]. In our study, although most participants reported a high level of participation throughout the development and implementation of the NSAP, some of them expressed that certain sectors were missing. These included the primary care and pharmaceutical sectors, as well as the media. Effective communication through targeted engagement to foster One Health collaboration is only possible with the involvement of all stakeholders, reinforcing the need for a diffusion of governance, as highlighted above. Some additional sectors that were mentioned by other studies for a balanced and practical view to achieve the NAP objectives included the finance, marine science, social science, and community health sectors [21,34].

Our study highlighted that awareness around the AMR issue was insufficient. For example, some participants were not familiar with AMRCO and AMRWG, while others were not familiar with the NSAP or the outcomes following its launch, although they are available online. This finding is similar to that in Hong Kong, where they found inadequate levels of engagement with non-government and private sectors [23]. Before improving public awareness, there is a need to ensure that all relevant stakeholders in the field are properly engaged. Lapses in engagement with professionals involved in AMR work could, in turn, impede the execution of key policies. One participant highlighted that knowledge on AMR was poor amongst reporters who play an important role in media communication for better public awareness. A 2019 systematic review of studies conducted on the media communication of antibiotics and AMR highlighted that the AMR issue was covered superficially in most media, with little focus on solutions to fight against AMR [41]. This simplistic coverage could be an issue as public understanding was not attained sufficiently. Educating those working in the media sector or developing a close collaboration between them and healthcare professionals or scientific experts is crucial to provide accurate and high-quality medical information to the public [41,42]. In another example, one participant mentioned that their environmental engineers are not well versed in the AMR concept. Although environmental AMR research has been increasing recently, improvements are needed to enhance its relevance and impact to health outcomes [43]. Proper engagement with environmental scientists and engineers is important to address knowledge gaps in AMR transmission from environments to humans and vice versa. This could be achieved through an interdisciplinary focus on areas, such as the identification of ‘hot spots’, like wastewater treatment plants, farms, and pharmaceutical effluents, as well as the characterisation of AMR within these ‘hot spots’ [44]. Participants also highlighted constraints in information sharing between agencies in the AMRWG. Our previous study in Singapore highlighted good transparency within the MOH and hospitals but not across other sectors [14]. Other similar studies largely discussed the need for better transparency to the public and not within the workgroups [22,24,45]. Transparency through information sharing is important as it facilitates stakeholder participation in policymaking, allows for external scrutiny, and demands for action, thereby instilling accountability [46,47]. In addition, it also fosters trust and helps advance implementation plans against AMR.

Defining a set of ideal indicators with targets and appropriate methods to measure these targets would help quantify the success of the plan over time. Study participants highlighted that although they tracked the progress of the implementation plans, there were no quantifiable targets in the NSAP because of the complexity of these indicators and the lack of baseline data. In our analysis of the ASEAN NAPs, we found that other than Brunei, Singapore, and Vietnam, all countries had some form of specific, measurable, and timebound indicators, with some of these objectives being specific quantitative targets for AMR and AMU in humans and animals [12]. Indicators were discussed in other similar qualitative studies as well, although the actual tracking of these indicators was not fully realised [22,24,35]. Participants further elaborated that with the current status of AMR efforts, it was intentional that the indicators in the NSAP were not too specific, to allow stakeholders to pick the targets that they thought were the most important and achievable. Participants also mentioned self-accountability among the stakeholders to accomplish what they promised, a feature which might be specific to the culture of Singapore [48]. A future expansion in surveillance and its integration across sectors would provide better baseline data that contribute towards the identification of appropriate and realistic risk-based targets to drive AMR control efforts [8]. In addition, there are plans to develop monitoring and evaluation indicators in the ASEAN region, following the launch of the ASEAN Strategic Plan [15].

Table 2 features a list of policy recommendations, developed based on our study results, recommendations from the participants, as well as findings from the literature [26,29,30,49,50,51].

To our knowledge, this is one of the first qualitative studies exploring the policy process and development of the NSAP of Singapore. The AMR governance framework that we developed previously allowed for a structured analysis. We were able to gather perspectives from a range of stakeholders in the human health, animal health, and environmental sectors. However, representation from the human health sector was much greater compared to that from the animal health and environment sectors. This was expected as the human health sector was much more involved in AMR-related activities. There were some limitations. Firstly, conducting the interviews virtually could have limited rapport building, which is crucial to enhance the quality of the interviews. Next, given the qualitative nature of the study, which adopted an interpretative approach that focused on the participants’ perspectives, there may be minor inaccuracies regarding some of the national programmes and structures. However, this was minimised to the best of our ability, following the member check process at the final stage of manuscript preparation where all participants were given the opportunity to make suggestions where appropriate. Third, the topic of equity was not discussed by our participants. This could have been downplayed because of the ubiquitous deployment of universal health coverage through a mixed healthcare financing system in Singapore. Fourth, findings from interviews are not easily generalizable to other countries, as this study was closely related to the Singapore context. However, the data would be useful for a comparative analysis of data from different countries that conducted their studies following the same methodology. Lastly, while we managed to elucidate the policy process and development of the NSAP, this paper is unable to detail the implementation processes of the NSAP. The latter will be covered in a complementary paper based on the same methodology and participant population.

## 5. Conclusions

Using an AMR governance framework, this paper examined the policy process and development of the NSAP in Singapore. While there was good governmental support and a dedicated secretariat to ensure good coordination across agencies, there was a need for more participation from certain sectors, better awareness and transparency, as well as the development of context-specific key indicators to track progress. Improvements in these areas will provide a more holistic One Health engagement for more effective planning and implementation of the NSAP.

## Figures and Tables

**Figure 2 antibiotics-12-01322-f002:**
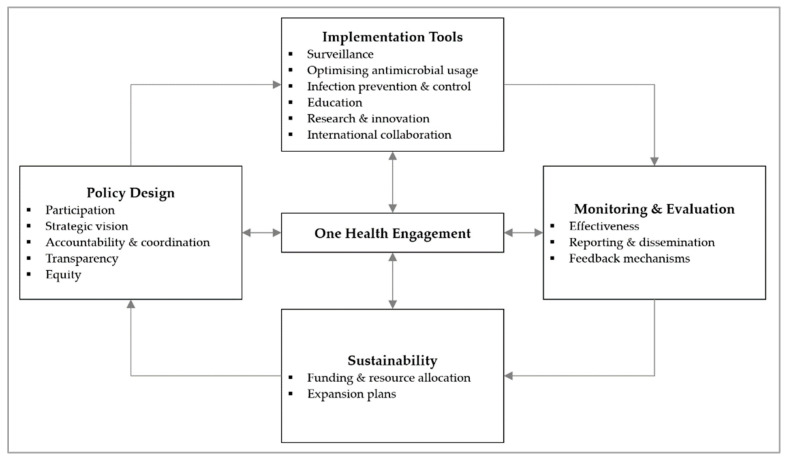
Governance framework for assessment of national action plans on antimicrobial resistance, adapted from Chua et al., 2021 [12]. Reuse is licensed under an open access Creative Commons CC BY 4.0 license.

**Table 1 antibiotics-12-01322-t001:** Summary of participants by type of institution and sector.

Type of Institution	Sector			Total
	Human Health	Animal Health	Environment	
Academia		1	1	2
Government agency	6	3	2	11
Hospital (public and private)	5			5
Industry	1			1
Primary care	1			1
Total	13	4	3	20

**Table 2 antibiotics-12-01322-t002:** Policy recommendations.

S/N	Policy Recommendations	Details
1	Appoint a dedicated inter-ministerial/cross-sectoral joint secretariat	A fully committed secretariat can support the function of the national One Health steering committee for effective cross-sectoral coordination.
2	Enhance engagement of all stakeholders at all levels	Education of stakeholders in positions of power can empower them to make better decisions and facilitate advocacy. Representation from stakeholders on the ground during policy discussions can help in development of relevant policies that are more palatable.
3	Increase participation from stakeholders beyond the One Health agencies involved in human health, animal health, and environment	Identifying and tapping networks of actors not yet fully enlisted can strengthen collective action to address antimicrobial resistance. These actors include but are not limited to those from media, finance, and social science, as well as the primary care and pharmaceutical sectors.
4	Better policy advocacy for increased awareness and sustained behavioural change	Multimodal advocacy efforts, including engagement through joint exercises and participatory processes can increase awareness, help guide the design of relevant interventions, and to improve stakeholder ownership.
5	Develop efficient data sharing systems to improve transparency	Transparency can improve trust and accountability. Robust and reliable data can support policy engagement, as well as monitoring and evaluating impact of interventions.
6	Set context specific indicators for the implementation plans	Indicators should be relevant, readily available, feasible to collect, and sensitive to changes. Baseline indicators provide a comparison to advise if the plans are on track or if adjustments must be made. Regular monitoring and evaluation of indicators help determine effectiveness of an intervention and provide evidence to inform policies.
7	Increase mobilisation of financial resources to support sustainability of implementation plans	Dedicated public and private sector investment across the One Health spectrum can provide sustainability in the interventions to tackle antimicrobial resistance.

## Data Availability

Not applicable.

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
