# Peer review of "A Qualitative Study on the Policy Process and Development of the National Action Plan on Antimicrobial Resistance in Singapore"

_antibiotics, 2023, doi:10.3390/antibiotics12081322_

Round 1

Reviewer 1 Report

I find This research on Policy Process and Development of

the National Action Plan on Antimicrobial Resistance to be interesting and of interest to a broad audiance. I believe it will add to the body of literature on the topic of microbial resistance. I also find the tables, figures and literature appropriate.

There is only a Minor comment for the authors, I would suggest putting the table 2 from discussion to results section. 

Author Response

Point 1: I find this research on Policy Process and Development of the National Action Plan on Antimicrobial Resistance to be interesting and of interest to a broad audience. I believe it will add to the body of literature on the topic of microbial resistance. I also find the tables, figures and literature appropriate.

Response 1: Many thanks for your kind comment as well as the time taken to review our article, we really appreciate it. We are also glad that you found the tables, figures, and literature appropriate! 

Point 2: There is only a Minor comment for the authors, I would suggest putting the table 2 from discussion to results section.

Response 2: Thank you very much for your suggestion!

We have also considered shifting table 2 to the results section. However, there are a few reasons why we think that it fits better in the discussion section in the end.

  1. The table of recommendations is derived from our study results and recommendations from the participants, as well as findings from the literature. In view of the latter, which is not part of the data collection process of the study, we find that it is not as appropriate to place the table in the results section.
  2. Some of the points in the table are elaborated in the discussion section. As such, we feel that the table fits better at the end of the discussion section as a summary to all the points that have been discussed.

We hope that this is acceptable to you. Otherwise, we are happy to update accordingly.

Reviewer 2 Report

This study offers valuable insights into the development of Singapore's NSAP on AMR, highlighting commendable aspects such as good coordination and governmental support. However, it also identifies areas needing improvement, such as limited participation from certain sectors, inadequate awareness, information sharing constraints, and the absence of ideal indicators for tracking progress. Addressing these shortcomings will enhance One Health engagement and promote more effective planning and implementation of the NSAP. 

The manuscript is well written with detailed methodology. However, some of the comments for further improvement of the draft are: 

Line 93: Please check the subheading S.No.

Line 228: Expand MOH

Line 240: IDI18, Human Health: Is this quote is not from Pharma Industry personnel?

Line 283: Elaborate the role of environmental engineers in detail in the discussion section.

Line 290: Educating reporters on public health issues is highly neglected in many regions. I appreciate to take up this point.

Is the gender distribution of the participants could have any effect on the study results? 

The discussion can be improved with special emphasis on the limitations of the present study.

Author Response

Point 1: This study offers valuable insights into the development of Singapore's NSAP on AMR, highlighting commendable aspects such as good coordination and governmental support. However, it also identifies areas needing improvement, such as limited participation from certain sectors, inadequate awareness, information sharing constraints, and the absence of ideal indicators for tracking progress. Addressing these shortcomings will enhance One Health engagement and promote more effective planning and implementation of the NSAP.

Response 1: Many thanks for your kind comment as well as the time taken to review our article, we really appreciate it. We hope that this article will generate more interest in AMR-related work across the One Health spectrum, especially in areas where it is currently not as advanced.

Point 2: The manuscript is well written with detailed methodology. However, some of the comments for further improvement of the draft are:

Response 2: Many thanks once again. We have addressed the comments point-by-point below. We hope that the edits are acceptable. Otherwise, we are happy to update accordingly.

Point 3: Line 93: Please check the subheading S.No.

Response 3: Thanks for this. We have deleted the numbering ‘1.2’, as we understand from previous experience with the editors that labels are not needed if there is only one subsection in the section.

Point 4: Line 228: Expand MOH

Response 4:  Thanks for the suggestion. We will keep ‘MOH’ in Line 229 in the abbreviated form as the first occurrence is at Line 67, ‘The workgroup is chaired by the representative from the Ministry of Health (MOH), which oversees human health.’.

Point 5: Line 240: IDI18, Human Health: Is this quote is not from Pharma Industry personnel?

Response 5: Thanks for the question. Yes, we have described that the participant is from the pharmaceutical sector in the text prior to the quote. However, for the labelling of the quotes, we have kept the categories to ‘Human Health’, ‘Animal Health’, and ‘Environment’.

We hope that this clarifies. Otherwise, we are happy to provide further details.

Point 6: Line 283: Elaborate the role of environmental engineers in detail in the discussion section.

Response 6: Thank you very much for this.

We have added a few lines in the discussion on the need for engagement of environmental scientists and engineers from Line 594 to 602. “In another example, one participant mentioned that their environmental engineers are not well versed in the AMR concept. Although environmental AMR research have been increasing recently, improvements are needed to enhance their relevance and impact to health outcomes. Proper engagement with environmental scientists and engineers is important to address knowledge gaps in AMR transmission from environments to humans and vice-versa. This could be achieved through an interdisciplinary focus on areas such as identification of ‘hot spots’ such as wastewater treatment plants, farms, and pharmaceutical effluents, as well as characterisation of AMR within these ‘hot spots’.”

Point 7: Line 290: Educating reporters on public health issues is highly neglected in many regions. I appreciate to take up this point.

Response 7:  Thank you very much for this. We agree with you on the need for better knowledge in the media sector.

We have added a few lines in the discussion from Line 586 to 594. ‘One participant highlighted that knowledge on AMR was poor amongst the reporters who play an important role of media communication for better public awareness. A 2019 systematic review of studies conducted on media communication of antibiotics and AMR, highlighted that the AMR issue was covered superficially in most media with little focus on solutions to fight against AMR. This simplistic coverage could be an issue as public understanding was not attained sufficiently. Educating those working in the media sector or developing a close collaboration between them and the healthcare professionals or scientific experts is crucial to provide accurate and high-quality medical information to the public.’

Point 8: Is the gender distribution of the participants could have any effect on the study results?

Response 8: Thank you for the comment. We had 11 female and male participants for this study.

We did not detect any theme related to gender issues in our study as none of the participants discussed this topic during their interviews.

However, we have also added an additional line to highlight the gender distribution of the participants in Line 116 – ‘Eleven participants are female while the rest are male.’

Point 9: The discussion can be improved with special emphasis on the limitations of the present study.

Response 9: Thank you very much for this. We have expanded the limitations of the study, highlighting two points on (1) the nature of the qualitative interpretative approach and (2) generalizability and applicability of the study.

Line 643 to 649, ‘Next, given the qualitative nature of the study which adopted an interpretative approach that focuses on the participants’ perspectives, there may be minor inaccuracies regarding some of the national programmes and structures. However, this was minimised to the best of our ability, following the member check process at the final stage of manuscript preparation where all participants were given the opportunity to make suggestions where appropriate.’

Line 651 to 655, ‘Fourth, findings from interviews are not easily generalizable to other countries, as this study was closely related to the Singapore context. However, the data would be useful for a comparative analysis of data from different countries who conducted their studies following the same methodology.’.

We hope that this is ok. Otherwise, we are happy to provide further details.

Reviewer 3 Report

This paper discusses an important issue, from a qualitative angle. The authors are skilled in the technique of designing and carrying out this neglected field of research, and the study is well designed, and flawlessly carried out.

The only addition I would suggest is an emphasis on the applicability of such a study in countries which are not as organized as Singapore.

Author Response

Point 1: This paper discusses an important issue, from a qualitative angle. The authors are skilled in the technique of designing and carrying out this neglected field of research, and the study is well designed, and flawlessly carried out.

Response 1: Many thanks for your kind comment as well as the time taken to review our article, we really appreciate it.

Point 2: The only addition I would suggest is an emphasis on the applicability of such a study in countries which are not as organized as Singapore.

Response 2: 

Thank you very much for the suggestion. Unfortunately, we feel that the findings from this study are not easily generalizable or applicable to other locations which may be less organized. However, this will be especially useful for comparison when we review data from different countries which used the same methodology to conduct the study in their own context. An example could be a comparison between a selection of high-income countries and low- and middle-income countries.

We have elaborated this in Line 651 to 655, ‘Fourth, findings from interviews are not easily generalizable to other countries, as this study was closely related to the Singapore context. However, the data would be useful for a comparative analysis of data from different countries who conducted their studies following the same methodology.’.

We hope that this clarifies. Otherwise, we are happy to provide further details.